# Crosstalk Between Liver and Extra-Liver Organs During Liver Regeneration in Mammals

**DOI:** 10.3390/cells14211691

**Published:** 2025-10-28

**Authors:** Andrey Elchaninov, Daria Bykova, Elena Gantsova, Polina Vishnyakova, Timur Fatkhudinov, Gennady Sukhikh

**Affiliations:** 1Laboratory of Growth and Development, Avtsyn Research Institute of Human Morphology of FSBI “Petrovsky National Research Centre of Surgery”, 3 Tsurupa Street, 117418 Moscow, Russia; dashasam@mail.ru (D.B.); gantsova@mail.ru (E.G.); tfat@yandex.ru (T.F.); 2Research Institute of Molecular and Cellular Medicine, Peoples’ Friendship University of Russia (RUDN University), 6 Miklukho-Maklaya Street, 117198 Moscow, Russia; vpa2002@mail.ru; 3Laboratory of Regenerative Medicine, Institute of Translational Medicine, National Medical Research Centre for Obstetrics, Gynecology and Perinatology Named After Academician V.I. Kulakov of Ministry of Healthcare of Russian Federation, 4 Oparina Street, 117997 Moscow, Russia; gt_sukhikh@bk.ru

**Keywords:** liver, regeneration, spleen, lung, kidney

## Abstract

Liver diseases are leading causes of death and disability worldwide. Therefore, the development of new treatments for liver disease is urgent. An integral part of the treatment of any liver disease is the process of regeneration of its parenchyma. However, the current approach to studying mammalian liver regeneration is one-sided, focusing solely on the liver itself and neglecting its interactions with other organs. This includes organs of the functional excretory system, such as the lungs and kidneys, which can secrete factors, including HGF, EGF, TGFβ, IL6, and others, into the bloodstream, which influence reparative processes in the liver. Only a few studies exist in this area. However, without a systematic approach to studying liver regeneration, it is impossible to develop new and effective methods to stimulate its regeneration. This review presents data on the interaction of the regenerating liver with the nervous, endocrine, and immune systems, as well as with the lungs, kidneys, and intestines.

## 1. Introduction

Liver diseases constitute a major cause of death globally [1]. A common manifestation of advanced chronic damage to the organ is cirrhosis, defined as pathological scarring of the liver, affecting 0.3–0.6% of the adult population according to various estimates [2]; the mortality over the first year post-diagnosis may reach 57% [1]. The extreme severity of advanced symptoms reflects the organ’s core systemic relevance.

In mammals, the liver is one of the few structures with pronounced self-repair capacity [3,4]. The regenerative potential of the liver constitutes an important biomedical focus as both a casualty and a counter to chronic diseases. Experimental models used in the field of liver regeneration include partial hepatectomy; hepatotoxic injury induced by carbon tetrachloride, CCl4, D-galactosamine, acetaminophen, thioacetamide, etc.; and transgenic animal lineages [5].

Most studies on liver regeneration involve one characteristic assumption. The regenerating liver is regarded as a self-sufficient autonomous organ endowed with vital components for the restoration of functional parenchyma. Meanwhile, in actual settings, the regeneration capacity of the liver is influenced by the concomitant liver failure and its deteriorating systemic consequences [6,7]. Particular examples include the hepatorenal syndrome amounting to functional renal failure in cirrhosis [8] and the hepatopulmonary syndrome with pathological changes to respiratory microcirculation [9,10,11]. These phenomena are consistent with the theory of functional systems by P.K. Anokhin [12], which regards the liver, the lungs, and the kidneys as an integrated excretory functional system; however, interactions between a regenerating liver and other excretory functionalities remain understudied. Several extraneous influences on regenerating liver so far considered in the literature include the autonomic nervous system, thyroid hormones, pancreatic insulin, and epidermal growth factor (EGF) secreted by salivary and intestinal glands [7].

Due to the vitality of various functions performed by the liver, a drastic change to its homeostasis through resection or toxin administration will profoundly affect the functional state of other organs; conversely, the condition of other organs affects the regeneration processes in the liver. This notion has been supported experimentally [13] but remains immature in terms of scientific rigor.

Importantly, the liver is the principal barrier separating the digestive tract contents from the internal milieu of the body. The portal blood carries the absorbed bacterial products, toxins, food antigens, etc., which pose a continuous immunological challenge jointly moderated by immune and non-immune elements of the liver. The hepatic immune microenvironments are overtly tolerant despite the abundance of innate immunity elements exemplified by resident Kupffer cells [14,15,16], and the connection of liver regeneration with immunity reactions is complex. We consider this issue in forthcoming sections (Figure 1).

## 2. Overview of Cellular and Molecular Mechanisms of Liver Regeneration

The liver is a parenchymatous organ comprising a complex ensemble of various cell types. The liver parenchyma is composed of hepatocytes organized into hepatic trabeculae. Hepatocytes account for over 80% of the liver’s mass [4]. Sinusoidal capillaries are located between the hepatic trabeculae, converging radially toward the central vein. This forms the classic lobule—the structural and functional unit of the liver.

The wall of the sinusoids is formed by endothelial cells, which account for approximately 20% of all liver cells [17]. In addition, resident liver macrophages—Kupffer cells—are found in the wall of the liver’s sinusoids [18]. Hepatic stellate cells—Ito cells, or perisinusoidal lipocytes—are located in the perisinusoidal space. They account for 10–15% of all liver cells [19]. Bile canaliculi, which lack their own walls, are located between the hepatocytes within the trabeculae. The bile duct’s own lining develops only at the level of the interlobular canals, which are lined with epithelial cells called cholangiocytes. A short section of intralobular bile duct with an epithelial wall is known as the canal of Hering [20]. It is thought to be the site of liver stem cells. Normally, in humans, rats, mice, and other animals, the lobule boundaries are not clearly defined due to the almost complete absence of connective tissue between the lobules. In addition to these cells, the liver also contains its own lymphocyte population, containing NK, NKT, and other subpopulations [21].

Molecular and cellular mechanisms of liver regeneration have been addressed profoundly; the main findings are as follows. Regeneration is triggered by a sharp decrease in the number of functionally active hepatocytes, which normally constitute more than 80% of the liver mass; the leading mechanism is mitotic proliferation of hepatocytes (in hepatectomy, those of the remnant lobes) [4]. The formation of new hepatocytes via transdifferentiation of the bile duct epithelial cells, cholangiocytes, is possible, but its relevance remains uncertain [22]. Non-parenchymal liver cells (endothelial, etc.) proliferate as well, with a minor contribution to the compensatory growth.

Experimentally, a sharp reduction in the number of viable hepatocytes can be modeled by either liver resection or massive cell killing by hepatotropic toxin exposure [23,24]. The two intervention types produce strikingly different histological pictures. Resections are ‘neat’ and convey no alternative changes to the tissue or signs of inflammatory infiltration in the remnant. Toxic exposures, by contrast, produce high numbers of hepatocytes dying or with signs of dystrophy and promote infiltration of the liver with granulocytes and monocytes. The nature of the impact (resection or toxin exposure) determines the repair scenario: the early onset of hepatocyte proliferation post-resection vs. delayed mode (after the damaging influence abates) in the case of toxic injury [23,24].

Hepatocytes are the first cells to enter the mitotic cycle after liver injury. Under normal conditions, almost all hepatocytes reside in G0, with mitotically dividing cells found at a frequency of 1:20,000, chiefly in the intermediate zone of the classical lobule (between the pericentral and periportal zones) [25]. Mitotic cycle reentry is thought to proceed in two stages: priming and G0/G1 transition. Through priming, supposedly orchestrated by TNFa and IL6 produced by liver macrophages, hepatocytes become more sensitive to mitogens. The G0/G1 transition is induced by several growth factors, primarily EGF and HGF; the latter is produced by endothelial cells of sinusoidal capillaries and stellate Ito cells of the liver. Inactive pro-HGF accumulates in the extracellular matrix (ECM) of the liver to be released and converted into the active form in the case of damage. Overall, damage-induced hepatocyte proliferation depends on multiple parameters, including the parenchymal loss volume, age, time of the day, etc., and is suppressed by TGFβ after regeneration [25,26].

Cholangiocytes, like hepatocytes, are mitotically dormant cells, i.e., their physiological proliferation rates are extremely low [27]. There are so-called small and large cholangiocytes, which contribute differentially depending on the inducing stimuli. Thus, common bile duct ligation stimulates large cholangiocytes to enter proliferation [28], while CCl4-induced acute injury is followed by small-cholangiocyte proliferation [29], and both small and large cholangiocytes proliferate after 70% liver resection [30]. Considering that proliferation of cholangiocytes is preceded by proliferation of endothelial cells of blood capillaries around bile ducts [31], elevated VEGF synthesis can have an early stimulatory effect on cholangiocyte proliferation [32]. Somatostatin has a blocking effect on cholangiocyte proliferation, although this only applies to large cholangiocytes [27,33]. A blocking effect on cholangiocyte proliferation has also been demonstrated for gastrin [34].

Proliferation of liver sinusoidal endothelial cells (LSECs), macrophages, and Ito cells does not contribute significantly to the compensatory growth. However, since these cell types produce various structural and regulatory molecules, their role in liver regeneration is essential.

Damage to the liver causes activation of Ito cells into a functional modality depending on the type of damage. Post-resection, Ito cells modulate the repair process at all stages by secreting a complex range of regulatory and structural molecules, including cytokines, growth factors, matrix metalloproteinases and their inhibitors, and ECM components [35,36]. At early stages of the process, Ito cells produce HGF, the principal mitogen for hepatocytes [7]. At later stages, Ito cells produce TGFβ, the major hepatocyte proliferation inhibitor also known to stimulate ECM production [37].

By contrast, toxic injuries propel Ito cells toward a myofibroblast-like phenotype; the activation is accompanied by a gradual decrease in the amount of vitamin A-containing lipid droplets, increased expression of a-SMA, and formation of cytoplasmic processes [38]. The activated Ito cells proliferate, move, and abundantly produce ECM components, primarily collagen type I. In chronic toxic liver injury, Ito cells abundantly produce tissue inhibitors of matrix metalloproteinases, which leads to excessive deposition of ECM and overall disruption of the architecture, amounting to fibrosis [39].

Sinusoidal endothelial cells of the liver constitute about 15–20% of its total cell count and less than 3% of its volume [17]. Physiologically, LSECs are characterized by low turnover rates [40]. In the event of liver damage, new LSECs are formed by proliferation and differentiation from either resident or bone marrow progenitors [41]. LSECs enter proliferation when stimulated by VEGF and FGF [41]. In toxic injury models, bone marrow progenitors are the main source of the new endothelial cells of the liver [42].

The liver has one of the largest macrophage populations in the body, heterogeneous in terms of cell origin. Three major lineages of liver macrophages include resident Kupffer cells, macrophages differentiated from peripheral blood monocytes, and liver capsule macrophages. Each lineage participates in liver repair to a varying extent depending on the type of damage [43,44].

At early stages of acute injury with acetaminophen or similar hepatotoxic agents, monocyte-derived macrophages increase in number, while Kupffer cells die in large numbers [45,46] to be restored via proliferation at later stages of the toxicity-induced inflammatory process [43,44].

The role of monocyte-derived macrophages in the post-resection recovery of the liver has long been a subject of controversy. Early studies emphasize the apparent lack of leukocyte infiltration from the blood to the residual organ [47]; however, the perspective was gradually modified by accumulating evidence. Monocytes capable of differentiation into macrophages were shown to migrate to the liver post-resection [48,49,50,51], although specific fates of these cells inside the remnant organ are debatable: some studies demonstrate their elimination, while others demonstrate their long-term survival in the liver [43,44].

## 3. The Influence of the Nervous System on Liver Regeneration

The conditioning influence of the nervous system on liver regeneration in mammals has been a long-standing subject of research. Experiments conducted in the mid-20th century showed more facile recovery of the liver mass after 70% resection in decorticated rats than in animals with an intact cerebral cortex [52].

The influence of the autonomic nervous system on liver regeneration has also been addressed experimentally [53,54,55]. Such findings include increased hepatocyte proliferation rates in remnant livers of vagotomized rats [56], partially supported by later evidence [57]. However, in other settings, vagotomy inhibited liver regeneration after partial hepatectomy in rats [58], and similar data were obtained in a mouse model of steatohepatitis [59]. One possible explanation of the inhibitory effect of vagotomy involves suppression of IL-22 synthesis in the remnant liver, which disrupts the production of hepatocyte proliferation agonists, notably STAT3 and anti-apoptotic factors. In a recent study, vagotomy-associated inhibition was neutralized by exogenous administration of IL-22 [60]. Excision of the hepatic branch of the vagus nerve (‘hepatic vagotomy’) also produced a delay in liver regeneration after partial hepatectomy in mice, especially in the early postoperative period marked by a significant decrease in Netrin-1, a key axonal growth and guidance protein. Knockout of the corresponding gene Ntn1 suppressed the synthesis of VEGF and HGF proteins, thereby inhibiting liver regeneration after partial hepatectomy. In the same genetically modified mouse model, exogenous Netrin-1 rescued HGF and VEGF synthesis and promoted a decrease in serum alanine and aspartate transaminase levels compared with the control group [61]. Another study associates the inhibitory effect of vagotomy on liver regeneration with decreased expression of Foxm1 in hepatocytes; noteworthy, the connection is mediated by liver macrophages [62]. The influence of the vagus nerve on liver regeneration can also be mediated by serotonin-producing cells of the small intestine: activation of the vagus nerve augments the release of serotonin, which supports liver regeneration [63].

Serotonin’s effects on liver regeneration are also exerted via the sympathetic nervous system. Enhanced binding of serotonin to its 5-HT2C receptors in the cortex and brainstem after liver resection boosts norepinephrine levels [64], thereby stimulating the synthesis of HGF and EGF, major mitogens for hepatocytes [7]. However, it should be remembered that the source of norepinephrine synthesis can also be the stellate cells of Ito [65,66].

Thus, the influence of the nervous system on liver regeneration is complex and multimodal. Despite the long-standing interest, the findings are often controversial, and specific mechanisms remain uncertain. The effects involve both the autonomic nervous system, notably vagal and sympathetic projections, and the neuroimmune and neuroendocrine interactions.

## 4. The Endocrine Influences

Endocrine influences on liver regeneration also have a long history of study [52]. A review by Abu Rmilah et al. (2020) [67] analyzed the effects of various hormones on liver regeneration as studied using human and animal cell cultures and rodent models. The analysis identified several hormones beneficial for the course of liver regeneration, including somatotropin, insulin, thyroid hormones, and norepinephrine, which stimulates the production of EGF and HGF, activates the Ras-Raf-MEK-ERK and PI3K-PKB-mTOR kinase cascades, and triggers the canonical Wnt/β-catenin pathway in liver cells [67].

Somatotropin, or growth hormone (GH), can also support liver regeneration by stimulating the release of growth factors (HGF and EGF) and activating the Ras-Raf-MEK-ERK and PI3K-PKB-mTOR kinase cascades. Most studies of GH in this context used knockout and transgenic mouse models. Reduced GH levels have been associated with increased severity of damage, whereas exogenous administration of GH stimulates EGFR expression and supports hepatocyte proliferation [67,68].

The stimulating effect of GH on hepatocyte proliferation is largely mediated by IGF-1 (insulin-like growth factor 1, a key signaling molecule of regeneration) produced in the liver. IGF-1 promotes hepatocyte proliferation and cell survival after hepatotoxic injury or partial hepatectomy by activating the intracellular signaling cascades of growth, notably the PI3K-Akt-mTOR and Ras-Raf-MEK-ERK pathways. Administration of GH to hypophysectomized rats (i.e., depleted of the endogenous GH) boosted HGF mRNA levels post-resection, leading to increased rates of hepatocyte proliferation compared with the control group [69].

The stimulating effect of insulin on liver regeneration is notable as well. Apart from its role in maintaining the energy balance and glucose metabolism of the regenerating liver, providing hepatocytes with nutrition essential for growth and recovery, insulin can promote hepatocyte proliferation. Insulin resistance can negatively affect liver regeneration by inhibiting hepatocyte growth and aggravating damage [67]. As demonstrated by Amaya et al. (2014) [70], insulin treatment triggers translocation of the insulin receptor to the nucleus in hepatocyte cultures. The mitogenic effects of insulin on hepatocytes are supposedly mediated by inositol-3-phosphate and Ca^2+^, as well as the PI3K-PKB-mTOR pathway [70].

Thyroid hormones T3 and T4 are also mitogens for hepatocytes. Pibiri et al. studied the influence of T3 on hepatocyte proliferation after partial hepatectomy in a rat model. Under the influence of T3, the peak of DNA synthesis (BrdU incorporation rates) in the remnant liver was reached much earlier than in the control group (respectively, 18 h and 24 h post-resection). Without affecting mRNA levels of key transcription factors (Ap-1, Nf-Κβ, Stat3; Emsa) and immediate early genes (Fos, Jun, Myc), T3 was shown to specifically activate E2F transcription factors, thereby accelerating the G1/S transition. The activities of cyclin-dependent kinases CDK2,4,6 and cyclin E were similar in both groups of the study [71]. The principal role of E2F was confirmed by Alisi et al. (2005); the team also demonstrated that T3 and T4 reduce the levels of p16 and p27, which act as cyclin-dependent kinase suppressors [72].

Experimental findings indicate that T4, like T3, is a potent pro-proliferative agent for hepatocytes. It stimulates the expression of E2F transcription factors and enhances the synthesis of cyclins and cyclin-dependent kinases (CDKs). Concurrently, T4 suppresses p16/p27 CDK inhibitors and p53/p73 tumor suppressors, undermining their inhibitory effects on CDKs and p21, activating Ras-ERK and β-catenin pathways, counteracting the oxidative stress and apoptotic signaling, and mitigating the effects of anti-proliferative TGF-β/SMAD cascades [67].

Overall, the concerted systemic action of hormones refines and coordinates the regenerative response after hepatotoxic injury or liver resection.

## 5. The Immunity Influences

Of all organs and structures of the immune system, the spleen is most closely connected with the liver. The connection, widely termed the ‘hepatosplenic axis’, is both anatomical, defined by the presence of a portal vein carrying cytokines and other regulatory molecules, and possibly also migratory cells, from the spleen to the liver, and functional, as both organs participate in immune defense, barrier function, and blood storage [73]. Pathophysiological indications of the hepatosplenic axis were initially observed in patients with liver cirrhosis and subsequently studied in animal models. The hepatosplenic axis has been studied in the context of cirrhosis-associated fibrous deposition, both clinically and in experimental models [74]. Hepatocyte decay products reach the spleen with circulation and stimulate red pulp macrophages to produce TGFβ collected into the portal flow. Of note, liver macrophages can also produce TGFβ and promote its production by Ito cells. Splenectomy has been shown to mitigate liver fibrosis [75]; the effect may involve reduced systemic (serum) levels of LIGHT protein (aka TNFSF14), which promotes a JNK kinase-dependent reduction in TGFβ synthesis by liver macrophages [76].

The spleen has been characterized as a source of leukocyte migration to the liver via the portal vein in a model of cirrhosis induced by chronic administration of CCl4. In a study by H. Jiang et al. (2021), splenectomy mitigated the leukocyte infiltration of the liver, thereby reducing the release of TNF-α with an ultimate protective effect on hepatocytes [76]. Signs of leukocyte exodus from the spleen were observed in a partial hepatectomy (70% liver resection) mouse model, with counts of CD115+ cells (monocytes) and resident F4/80+ cells in the spleen decreasing [77]. Similar data were obtained in a liver resection model, where splenectomy reduced the rates of monocyte, lymphocyte, and granulocyte migration to the liver [49].

The clinical benefit of splenectomy in liver fibrosis is consistent with the splenectomy-induced increase in counts of pro-regenerative CD300E+ liver macrophages revealed by single-cell RNA sequencing. CD300E+ macrophages have been shown to stimulate hepatocyte proliferation by secreting NAMPT, a potent cytokine [78]. Apart from macrophages, the beneficial effect of splenectomy can be mediated by lymphocytes. A splenectomy-induced increase in circulating CD8+ cell counts and a concomitant decrease in the CD4+/CD8+ ratio in patients with liver cirrhosis have been associated with significant deceleration of fibrosis progression and improved anti-tumor immunity [79].

The beneficial effect of splenectomy on repair processes in the liver has also been demonstrated in liver resection models. IL10 is known to suppress the excessive pro-inflammatory response during early phases of liver regeneration post-resection by partially inhibiting pro-proliferative STAT3 signaling in hepatocytes [80]. In hepatectomized experimental animals with genetically inactivated Il10, the enhanced pro-inflammatory signaling promoted STAT3 cascade hyperactivation and ultimately increased the rates of hepatocyte proliferation [80]. Stimulation of hepatocyte proliferation is observed even in the intact liver after splenectomy [81].

Accordingly, removal of the spleen as an additional source of IL10 is likely to enhance hepatocyte proliferation. Still, several experimental studies demonstrate an inhibiting effect of splenectomy on liver regeneration post-resection. The degree of such inhibition observed by A.G. Babaeva et al. (1989) was independent of the time lapse between splenectomy and liver resection [82]. The findings can be explained as follows.

HGF is the chief mitogen for hepatocytes [83], and insufficient levels of this factor may negatively affect the rates of liver regeneration. Hepatectomy has been shown to stimulate *Hgf* expression not only in the liver itself but also in the spleen [84]. The impact of the spleen as a source of HGF for liver regeneration after hepatectomy has not been thoroughly addressed.

As shown in experiments with Il17a knockout mice, IL17A produced by CD4+ T cells of the spleen stimulates hepatocyte proliferation in regenerating liver. Moreover, the counts of IL17A-producing CD4+ T cells in the spleen are boosted after partial hepatectomy [85]. Of note, CD4+ T cells of the spleen not only produce IL17A for systemic delivery, but can also directly migrate to the regenerating liver.

Most importantly, splenectomy breaks the supply of splenic regulatory molecules to the regenerating liver. Our own data demonstrate a significant increase in expression of protease inhibitor-encoding genes *Serpina3n*, *Stfa2,* and *Stfa2l1* within the spleen after liver resection [50]. Protease inhibitors can exert anti-inflammatory effects, and a decline in their synthesis can negatively affect repair processes in various organs [86,87].

The influence of other organs and structures of the immune system on liver regeneration remains largely obscure. There is very limited evidence on the effect of thymectomy on liver regeneration. Removal of the thymus can delay liver regeneration in mice after partial hepatectomy, regardless of age, by decreasing the rate of hepatocyte proliferation [82,88]. Regeneration of the liver in T cell-depleted animals can be undermined by decreased production of IL6, which supports the transition of hepatocytes into the mitotic cycle [89]. On the contrary, activation of killer T cells in old mice can improve hepatocyte proliferation after partial hepatectomy, probably through induction of TNFα synthesis [90]. A classic study demonstrates that inhibition of liver regeneration in mice by thymectomy occurs only if the thymus is removed in the early postnatal period [91]. The overall confusion in the field probably reflects the diverse ages of animals used in different settings, as well as different time lapses between the surgeries.

Thus, the hepatosplenic axis orchestrates a complex interaction network of cells and regulatory molecules that either stimulate or inhibit the regenerative processes in the liver. Further understanding of the interplay between immunity and liver regeneration will require more refined and dedicated experimental settings accounting for age, immune status, and the nature of damage to the liver.

## 6. The Excretory Functional System

The excretory functional system comprises organs that protect the constancy of the internal environment of the body: the liver, the kidneys, the lungs, the skin, and the intestine [92]. Mutual influences between these organs under conditions of liver resection remain understudied.

Most findings in this area concern the altered circulation of bile acids between the liver and the intestine. The post-resection increase in bile acid influx per hepatocyte activates nuclear receptors, notably FXR, and associated signaling pathways that stimulate compensatory growth. With complete recovery of the liver mass, the influx of bile acids per hepatocyte returns to physiological values, nuclear receptor signaling ceases, and the emergency transcriptomic signatures return to basal levels [93]. Despite the obviously important role of bile acid cycling in regulating compensatory liver growth, its contribution to the production of principal regulatory molecules involved in the process remains obscure. A study by Ji et al. describes a connection between hepatointestinal cycling of bile acids with FGF-15 and the Hippo/Yap signaling pathway [94]. Another study demonstrates a negative effect of the colon on liver regeneration in a rat model. Colectomy performed in combination with 70% liver resection promoted a significant decrease in serum transaminase levels, an increase in HGF and TGFα content, and acceleration of liver regeneration compared to the control group (70% liver resection; no colectomy) [95].

Clinically, liver resections are often complicated by respiratory distress syndrome [96,97,98] and pneumonia [99]. Hepatorenal syndrome has long been acknowledged in pathophysiology; the term relates to severe functional acute renal failure, the impairment of kidney function as a consequence of liver damage, including post-resection. Despite the clinical relevance, pathogenetic mechanisms of hepatorenal syndrome remain uncertain [8].

As demonstrated in early experiments, 70% liver resection boosts *Hgf* expression in the lungs and in the kidneys [84,100]. The mechanism of HGF synthesis induction in the lungs and in the kidneys is probably similar to that described for the liver. After liver resection, blood levels of bacterial lipopolysaccharide (endotoxin) sharply increase, promoting activation of liver macrophages, which start to produce IL6 [7], an inducer of *Hgf* expression in Ito cells [101]. The high endotoxin levels may also activate resident macrophages in other organs, boosting IL6 production to activate HGF synthesis, probably by macrophages themselves and also by endothelial cells [84,100].

This assumption is consistent with our previous data on increased expression of not only *Hgf*, but also of *Il1b*, *Il6*, *Il10*, and *Tnfa* in the lungs and in the kidneys after subtotal liver resection in rats. Increased counts of pulmonary macrophages 24 h and 48 h post-resection also indicate their activation [102,103]. In the same setting, lung tissues revealed a gradual increase in the HGF protein content post-resection, while renal samples revealed a sharp increase in HGF content until 6 h post-resection, followed by a sharp decrease, probably indicating its release into the bloodstream [102,103].

A notable feature of liver regeneration after subtotal hepatectomy in the rat model is a temporary block in hepatocyte proliferation, the resolution of which may demand extra HGF, possibly involving synthetic capacities outside the liver [100,102,103]. The gradual compensation for acute liver failure during regeneration coincides with a reduction in demand for HGF and cessation of its synthesis in the lungs and kidneys. As the supply of circulating HGF to the liver decreases and its levels decline, HGF gene expression and protein synthesis are enhanced in the liver itself. The HGF levels of the liver after subtotal hepatectomy in rats are fully replenished by day 10 post-resection [104].

The role of cytokines and growth factors produced outside the liver post-resection depends on yields and rates of their synthesis, and whether they act locally or distantly (i.e., are delivered by circulation). The locally increased rates of synthesis of particular regulatory molecules in the lungs and in the kidneys may represent the liver failure-associated damage to these organs and have no influence on liver regeneration.

If the experimental data indirectly indicate that liver resection impairs tissue homeostasis in the lungs and in the kidneys, then how exactly the condition of these organs affects liver regeneration is unclear. Two decades ago, liver resection was shown to boost the expression of *Hgf* gene in the lungs and in the kidneys [84,105]. The induction of HGF synthesis in the lungs, the kidneys, and the spleen may involve mechanisms similar to those described for the liver. After liver resection in rodents, plasma endotoxin levels increase as a direct consequence of deficient clearance and trigger macrophage activation. The activated macrophages produce IL6 [7], a potent inducer of *Hgf* expression in Ito cells of the liver [101]. It is possible that the high endotoxin levels activate macrophages not only in the liver, but also in other organs, boosting local IL6 synthesis and augmenting HGF production by resident macrophages and endothelial cells in the lungs, the kidneys, and the spleen [84,100].

## 7. Conclusions

Over their long history, liver regeneration studies accrued considerable evidence of the participation of other organs and structures, primarily that of the nervous, endocrine, and immune systems. In terms of volume and depth, such evidence is substantially inferior to the knowledge of internal mechanisms of liver regeneration; thus, studies concerning the relationship of liver regeneration with the lungs and the kidneys are few in number. However, this line of research is extremely important from the standpoints of both basic science, studying the fundamental mechanisms of regeneration regulation in mammals, and clinical medicine. The applied significance is underscored by severe dysfunctions of other organs, primarily the lungs and the kidneys, associated with liver failure. At the same time, the lungs and kidneys can provide additional sources of biologically active substances, which should not be neglected when developing new methods for stimulating repair processes.

Further research is needed in this area. It is necessary to elucidate the contribution of biologically active substances secreted by other organs after liver injury to liver regeneration. Furthermore, it is necessary to establish the underlying mechanisms of interorgan interactions. The contribution of the nervous and endocrine systems is fairly well understood. However, the macrophage system, as well as the stellate cell system found in the liver, lungs, and pancreas, may also be involved in crosstalk between the liver and extra-liver organs.

## Figures and Tables

**Figure 1 cells-14-01691-f001:**
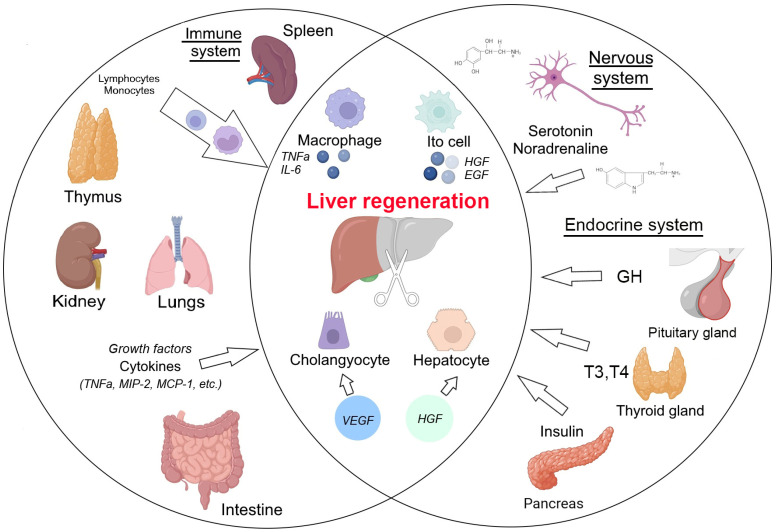
Crosstalk between liver and extra-liver organs during liver regeneration in mammals. GH—growth hormone, T3,T4—thyroid hormones—thyroxine (T4) and triiodothyronine (T3).

## Data Availability

No new data were created or analyzed in this study. Data sharing is not applicable to this article.

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
