# Peer review of "Crosstalk Between Liver and Extra-Liver Organs During Liver Regeneration in Mammals"

_cells, 2025, doi:10.3390/cells14211691_

Round 1

Reviewer 1 Report

Comments and Suggestions for Authors

The authors have generated a large document that covers most aspects of liver regeneration and relevant connections with the contribution of other organs, including nervous system spleen, lungs and kidney to the regenerative process. It will be a useful review for finding many important references to all the work that has been done in tis area.

Some points that the authors should address are as follows:

  1. The word "excretory", used in many parts of the paper, should be replaced by the word "secretory".
  2. There is much speculation on the role of spleen in the regenerative process, but rather few data on a meaningful contribution from spleen to the liver during regeneration. That section should shortened.
  3. Norepinephrine is mentioned in the endocrine contribution section of the review. Actually norepinephrine can be produced by the stellate cells of the liver and rises early after hepatectomy. Norepinephrine also enhances the effects of HGF and EGF. References to these process is shown in PMID 2982212, 2545731.
  4. The first sentence of the abstract should be modified by removing the word "the".

Author Response

Dear editors and reviewers!

Thank you for your attention to our work and valuable comments.   We would like to respond to them one by one.

  1. The word "excretory", used in many parts of the paper, should be replaced by the word "secretory".

Thanks for your comment. We have identified the liver, lungs, and kidneys as organs of the excretory system (https://en.wikipedia.org/wiki/Excretory_system). These organs are capable of secreting biologically active substances; however, after liver resection, the lungs and kidneys are additionally burdened with the removal of metabolic products. This does not preclude the ability of, for example, the lungs and kidneys to secrete biologically active substances into the blood.

2. There is much speculation on the role of spleen in the regenerative process, but rather few data on a meaningful contribution from spleen to the liver during regeneration. That section should shortened.

Thanks for the recommendation. The section has been shortened.

3. Norepinephrine is mentioned in the endocrine contribution section of the review. Actually norepinephrine can be produced by the stellate cells of the liver and rises early after hepatectomy. Norepinephrine also enhances the effects of HGF and EGF. References to these process is shown in PMID 2982212, 2545731.

Thanks for the recommendation. This information has been added to the text of the article.

4. The first sentence of the abstract should be modified by removing the word "the".

 Thanks for the recommendation. The text has been corrected.

Reviewer 2 Report

Comments and Suggestions for Authors

The manuscript titled ‘CROSSTALK BETWEEN LIVER AND EXTRA-LIVER ORGANS DURING LIVER REGENERATION IN MAMMALS’ by Elchaninov et al., provides us with a summary on crosstalk between liver and other organs. Based on the abstract and introduction, it appears that the authors intended to review how this crosstalk influences liver regeneration. However, the content is broader, including the effects of liver damage on other organs, which makes the paper less concise and focused.

  1. I suggest that the authors focus their review on the stated aim.
  2. Section 2 “Overview of cellular and molecular mechanisms of liver regeneration” is difficult to follow. It would be helpful to provide background information on liver composition and the physiological roles of different cell types (e.g., cholangiocytes, Ito cells—preferably referred to as hepatic stellate cells [HSC]) before describing their roles in regeneration.
  3. The first half of Section 5, “The Immunity Influences,” mainly discusses how liver damage (e.g., hepatocyte decay and liver fibrosis) affects the spleen rather than how these changes influence liver regeneration. To maintain focus, this part could be removed.
  4. Section 6, “The Excretory Functional System,” contains extensive discussion on the effects of liver resection on the lung and kidney rather than on how these organs affect liver regeneration. Restructuring this section to emphasize the main points would improve clarity.
  5. Finally, the conclusion would be clearer if it focused on the impact of organ crosstalk on liver regeneration.

Other small points

  1. Several references have no journals’ name, e.g. No. 8, 34, 47, 79 and 80.
  2. In line 88, it states that hepatocytes ‘normally constitute more than 80% of the liver mass’. I am not sure where the >80% come from as it is not in the reference cited.
Comments on the Quality of English Language

The English is fine overall. However, it could be improved for clarity.

Author Response

Dear editors and reviewers!

Thank you for your attention to our work and valuable comments.   We would like to respond to them one by one.

  1. I suggest that the authors focus their review on the stated aim.

Thanks for the recommendation. We tried to follow it.

2. Section 2 “Overview of cellular and molecular mechanisms of liver regeneration” is difficult to follow. It would be helpful to provide background information on liver composition and the physiological roles of different cell types (e.g., cholangiocytes, Ito cells—preferably referred to as hepatic stellate cells [HSC]) before describing their roles in regeneration.

The liver is a parenchymatous organ comprising a complex ensemble of various cell types. The liver parenchyma is composed of hepatocytes organized into hepatic trabeculae. Hepatocytes account for over 80% of the liver's mass [1]. Sinusoidal capillaries are located between the hepatic trabeculae, converging radially toward the central vein. This forms the classic lobule—the structural and functional unit of the liver.

The wall of the sinusoids is formed by endothelial cells, which account for approximately 20% of all liver cells [2]. In addition, resident liver macrophages—Kupffer cells—are found in the wall of the liver's sinusoids [3]. Hepatic stellate cells—Ito cells, or perisinusoidal lipocytes—are located in the perisinusoidal space. They account for 10-15% of all liver cells [4]. bile canaliculi, which lack their own walls, are located between the hepatocytes within the trabeculae. The bile duct's own lining develops only at the level of the interlobular canals, which are lined with epithelial cells called cholangiocytes. A short section of intralobular bile duct with an epithelial wall is known as the canal of Hering [5]. It is thought to be the site of liver stem cells. Normally, in humans, rats, mice, and other animals, the lobule boundaries are not clearly defined due to the almost complete absence of connective tissue between the lobules. In addition to these cells, the liver also contains its own lymphocyte population, containing NK, NKT, and other subpopulations [6].

  1. Michalopoulos GK, Bhushan B. Liver regeneration: biological and pathological mechanisms and implications. Nat Rev Gastroenterol Hepatol [Internet]. 2021 [cited 2021 Jun 9];18:40–55. Available from: https://pubmed.ncbi.nlm.nih.gov/32764740/
  2. Poisson J, Lemoinne S, Boulanger C, Durand F, Moreau R, Valla D, et al. Liver sinusoidal endothelial cells: Physiology and role in liver diseases. J Hepatol [Internet]. 2017 [cited 2019 Oct 20];66:212–27. Available from: https://linkinghub.elsevier.com/retrieve/pii/S0168827816303336
  3. Elchaninov A, Vishnyakova P, Menyailo E, Sukhikh G, Fatkhudinov T. An Eye on Kupffer Cells: Development, Phenotype and the Macrophage Niche. Int J Mol Sci 2022, Vol 23, Page 9868 [Internet]. 2022 [cited 2023 Apr 20];23:9868. Available from: https://www.mdpi.com/1422-0067/23/17/9868/htm
  4. Hellerbrand C. Hepatic stellate cells--the pericytes in the liver. Pflugers Arch [Internet]. 2013 [cited 2019 Oct 20];465:775–8. Available from: http://www.ncbi.nlm.nih.gov/pubmed/23292551
  5. Kordes C, Häussinger D. Hepatic stem cell niches. J Clin Invest [Internet]. 2013 [cited 2019 Oct 25];123:1874–80. Available from: http://www.ncbi.nlm.nih.gov/pubmed/23635785
  6. Wen Y. The Role of Immune Cells in Liver Regeneration. Livers 2023, Vol 3, Pages 383-396 [Internet]. 2023 [cited 2025 Oct 20];3:383–96. Available from: https://www.mdpi.com/2673-4389/3/3/29/htm

3. The first half of Section 5, “The Immunity Influences,” mainly discusses how liver damage (e.g., hepatocyte decay and liver fibrosis) affects the spleen rather than how these changes influence liver regeneration. To maintain focus, this part could be removed.

Thanks for the recommendation. The text has been corrected.

4. Section 6, “The Excretory Functional System,” contains extensive discussion on the effects of liver resection on the lung and kidney rather than on how these organs affect liver regeneration. Restructuring this section to emphasize the main points would improve clarity.

Thanks for the recommendation. The text has been corrected.

5. Finally, the conclusion would be clearer if it focused on the impact of organ crosstalk on liver regeneration.

Thanks for the recommendation. The text has been corrected.

Other small points

6. Several references have no journals’ name, e.g. No. 8, 34, 47, 79 and 80.

Thanks for your comment. The references have been corrected.

7. In line 88, it states that hepatocytes ‘normally constitute more than 80% of the liver mass’. I am not sure where the >80% come from as it is not in the reference cited.

Thanks for your comment. At the end of this sentence is a link to the article (https://www.nature.com/articles/s41575-020-0342-4). In this article, in the section «Regenerative activities of hepatocytes Intracellular events», data are presented on the proportion of liver mass occupied by hepatocytes.

Round 2

Reviewer 2 Report

Comments and Suggestions for Authors

The authors have taken the comments into account.

Comments on the Quality of English Language

The English is fine overall. However, it could be improved for clarity.